# The Effect of Heterogeneity and Leadership on Innovation Performance: Evidence from University Research Teams in China

**Shufang Huang** [1,*] **, Jin Chen** [2]**, Liang Mei** [3] **and Weiqiao Mo** [1,*]

1   School of Public Administration, Zhejiang University of Finance and Economics, Hangzhou 310018, China
2   School of Economics and Management, Tsinghua University, Beijing 100084, China
3   National School of Development, Peking University, Beijing 100871, China
*   Correspondence: hshf16@zufe.edu.cn (S.H.); wqmo214@163.com (W.M.)

**Abstract:** Interdisciplinary cooperation is an important way to achieve scientific innovation breakthrough. Currently, great scientific innovation often occurs in interdisciplinary areas. However, they still face challenges in relation to theoretical support and strategic choices. This paper identifies the extent to which interdisciplinary cooperation-induced heterogeneity affects team innovation performance in Chinese universities. The questionnaire survey is employed in this study and the samples selection covers a wide range of multidisciplinary or interdisciplinary collaboration. This study used Poisson regression analysis to create a new method to evaluate innovation performance. Then, the relationship between team heterogeneity and innovation performance was examined and the moderating role of transformational leadership was also introduced. The empirical results show that three independent variables (disciplinary heterogeneity, cognitive heterogeneity, and organisational heterogeneity) all had a significant positive effect on the team innovation performance. Transformational leadership has a significant positive effect on cognitive heterogeneity and innovation performance, but moderating effects did not appear to be seen in the other two relationships. Our study contributes to a deeper understanding of the value of interdisciplinary research collaboration.

**Keywords:** interdisciplinary cooperation; innovation performance; team heterogeneity; transformational leadership; China; university

## 1. Introduction

The Needham Puzzle [1] in relation to China's technological innovation remains unanswered. Recent events involving the Chinese telecommunications company ZTE have exposed China's problems in relation to independent innovation in the field of telecommunications. In addition to telecommunications technology, China lacks core technologies in areas such as new drug creation, fine chemicals, oil drilling, computer hardware and software, and blood diagnostic equipment. To overcome this situation, the Chinese government is striving to improve innovation capability, and thus the academic community needs to identify the key factors that promote innovation. Based on relevant theoretical and historical experience, interdisciplinary cooperation is seen as an important way to achieve innovation breakthroughs [2–6]. Thus, this constitutes the focus of this study.

Scholars hold differing views on the impact of heterogeneity on innovation performance [7]. Interdisciplinary research has been conducted from various perspectives based on a range of theories, but has mainly focused on resource-based theory [8,9], social cognitive theory [10] and social identity theory [11]. The resource-based view and social identity theory both argue that the heterogeneity

generated by interdisciplinary interaction promotes innovation performance among the participants, while social cognitive theory states that heterogeneity leads to cognitive barriers between the participants, and thus hinders innovation. Thus, to date, the results of studies on the impact of interdisciplinary teamwork on innovation performance have been inconsistent. This study addresses this issue, and confirms its theoretical significance.

In relation to interdisciplinary understanding, previous studies have mostly focused on the physical level, while failing to analyze psychological factors such as cognition, which is an important dimension in interdisciplinary research [12,13]. Therefore, this study introduces the cognitive dimension to the study of interdisciplinary team heterogeneity by dividing team heterogeneity into three dimensions: disciplinary heterogeneity, cognitive heterogeneity, and organisational heterogeneity.

This study analyzes the relationship between team heterogeneity and innovation performance in 255 university research teams involved in interdisciplinary cooperation. In addition, to examine the influence of leadership on the above relationship in the Chinese context, we introduced transformational leadership as a moderator. The results showed that cognitive heterogeneity has the most significant impact on innovation performance. Disciplinary heterogeneity, cognitive heterogeneity, and organisational heterogeneity all contribute significantly to the overall performance of innovation teams in colleges and universities. We believe that the positive effects generated in accordance with resource-based theory and social identity theory are more influential than the negative effects of team conflict based on social cognitive theory. Of the three types of heterogeneity, cognitive heterogeneity has the greatest impact on innovation performance, followed by disciplinary heterogeneity and organisational heterogeneity. The results outline the role of transformational leadership in modulating the relationship between heterogeneity and performance.

This study makes several contributions to both theory and practice in this field. First, in terms of the measurement of cognitive heterogeneity variable, the measurement of team members using scales, rather than using personal background characteristics (such as team tenure, education level, and economic level) as surrogate variables, serves to overcome deviations in the measurement of cognitive heterogeneity to some extent. Second, this study uses Poisson regression analysis to convert traditional innovation performance items into numerical variables, which helps to identify the essence of the innovation. Third, the results of this study provide practical guidance for strategic decision-making and interdisciplinary cooperation in relation to innovation in Chinese universities. The practical innovation process should pay particular attention to the impact of team members' cognitive differences on teams' innovation performance. The management process should aim to take advantage of the benefits of heterogeneity while avoiding its potentially negative effects.

In this article, Section 2 reviews the theoretical foundations of heterogeneity, transformational leadership, and innovation. Section 3 describes the material and methods that are used in this study. Section 4 displays the results of the relationship between heterogeneity and innovation performance. Section 5 discusses and concludes how to choose heterogeneous partners during the process of interdisciplinary cooperation, thus implement interdisciplinary theories and managerial implications.

## 2. Theoretical Foundations and Hypothesis Development

### 2.1. Interdisciplinary Heterogeneity and Innovation

Scholars hold differing perspectives on the impact of interdisciplinary collaborative research with heterogeneous partners on innovation performance [7]. One group of scholars believes that cooperation among heterogeneous resources from different disciplines can help to build a sustainable competitive advantage in organisations [14]. This view is mainly based on the resource-based view. Another group of scholars believes that the heterogeneity of interdisciplinary teams involved in the innovation process leads to cognitive and affective conflicts, and that the heterogeneity of team members in terms of their professional background and knowledge may lead to differences in cognition, thereby triggering conflicts in relation to thinking and ways of doing things. These conflicts may

affect the team's innovation performance [10]. This view is mainly based on social cognitive theory. Scholars subscribing to this view have developed a further understanding that, although cooperation between interdisciplinary teams can indeed lead to conflict among members, this conflict may occur at some point in a long-term process of cooperation, and once the conflict is resolved, the understanding and inclusiveness of the members are strengthened. Team cohesion increases, and thus the team's innovation capacity is enhanced.

This view is based primarily on social identity theory [15]. In summary, the impact of heterogeneity on a team's innovation performance may be either beneficial or harmful, and this may depend on the context of analysis [16]. In general, when the task is relatively complex and the problem is unconventional, a heterogeneous team is more likely to find a solution [17].

Currently, the viewpoint emphasizing the positive impact of heterogeneity is predominant. In line with the objective of this study, the sample research team engaged in complex scientific exploration. Therefore, they were faced with relatively difficult problems, i.e., the system was more complex. In this case, teams from different disciplines worked together, even though this may have led to conflicts between cognitively heterogeneous individuals. Most researchers involved in innovation have lofty ideals and aspirations; thus, even if they experience some degree of cognitive conflict, they will be able to achieve the original scientific aims of the project. In addition, researchers are generally talented individuals who are highly capable of learning new skills, and thus will soon learn how to communicate effectively with other researchers with different cognitive qualities.

Therefore, we assume that in relation to the university's innovation team, the positive effect of discipline heterogeneity is greater than the negative effect of any conflict it might create. To summarize, we propose the following hypothesis:

**Hypothesis 1.** *Disciplinary heterogeneity positively influences innovation performance.*

### 2.2. Cognitive Heterogeneity and Innovation

The impact of cognition on innovation is based on the emotions of scientists [13]. Science and mood are usually in juxtaposition because people generally think that science is rational, while emotions are irrational, and thus unconducive to scientific research [18,19]. However, studies have shown that there is a positive correlation between emotion and scientific research. Moreover, this correlation is closely related to the nature of the subject, in particular, the degree of "softness" or "hardness" of the subject [20,21]. The degree of softness or hardness of various disciplines and the influence of the emotions of the scientists can be explained by the theory of "similar absorption." In short, individuals with similar emotional makeups are more likely to appreciate and agree with each other.

Therefore, cooperation between heterogeneous scholars may be seen as contrary to this principle, thereby affecting the likelihood of exchange and sharing of information among the collaborators, which will have a negative impact on innovation. Second, the impact of cognitive heterogeneity on innovation may also be based on differences in values. Collaborators with cognitive heterogeneity often display a variety of beliefs and values that can lead to conflict in decision-making [22,23]. Of course, some scholars believe that heterogeneity can expand the knowledge and vision of the various members of the group, inspiring team members to generate more innovative ideas. Thus, a moderate degree of cognitive heterogeneity is beneficial to team innovation, but the impact depends on cross-discipline sharing of expertise, the division of specialization across experts, and the degree of mutual understanding [24,25].

Lavy, Bareli, and Ein-Dor (2015) [26] examined the relationship between team cohesion and team function in the context of research team heterogeneity and found that cognitive heterogeneity is positively correlated with innovation performance when the level of team cohesion is high.

Based on a comprehensive literature review, combined with the specific context of innovation within the university, we believe that, although cognitive heterogeneity may lead to conflict between

scientists in terms of values and emotions, a heterogeneous team of scientists can cooperate to build a high level of team cohesion. To summarize, we propose the following hypothesis:

**Hypothesis 2.** *Cognitive heterogeneity positively influences innovation performance.*

*2.3. Organisational Heterogeneity and Innovation*

In the case being examined, there is heterogeneity between the organisation where the ontology team is located and the organisation where the research team is located, which has an impact on innovation. For the study of organisational heterogeneity, relative cognitive heterogeneity should be mature. Scholars generally believe that heterogeneous organisations can work together to improve the quality of innovation [27]. Following an in-depth survey of scientific organisations in Britain, France, Germany, and the United States in the 20th century, Hollingsworth (2007) [28] pointed out that the main organisational features that promote scientific discovery and innovation can be summarized as follows: (1) greater scientific diversity; (2) the recruitment of scientists with diverse abilities; (3) frequent and close communication and interaction among scientists from different fields; (4) an environment with a global vision, the ability to integrate scientific diversity, a strategic vision, the ability to nurture high-quality scientific research, and leaders who have the ability to obtain funding to achieve organisational goals; and, (5) a flexible and autonomous system environment. Conversely, the main factors inhibiting discovery and innovation are as follows: (1) clear organisational boundaries and sectoral divisions; (2) hierarchical levels of entitlement; (3) excessive bureaucracy, with extremely rigid rules and procedures; and, (4) excessive diversity, which may prevent effective communication between collaborators in different fields. Therefore, the nature of the organisation is closely related to innovation performance.

Organisations can be divided into profit-seeking and non-profit organisations. Profit-seeking organisations mainly include private-sector enterprises, while non-profit organisations include universities, research institutes, and government agencies [29]. Scholars found that organisational heterogeneity had a significant positive effect on innovation performance [30]. In summary, the impact of organisational heterogeneity on innovation performance is unclear, but we have taken into account the fact that China has accumulated a lot of experience in terms of cooperation in relation to research and development activities, and has demonstrated a willingness to share information, and so the likelihood of experiencing misunderstanding and conflict through heterogeneity has gradually declined. As a result, although partnerships between teams from different organisations may have a negative impact, the advantages of the heterogeneous resources and knowledge that collaboration across organisations brings are more pronounced. Thus, we propose the following hypothesis:

**Hypothesis 3.** *Organisational heterogeneity positively affects innovation performance.*

*2.4. The Moderating Role of Transformational Leadership*

Based on the theory of intrinsic motivation, transformational leadership uses work design to promote teamwork among team members. As a result, the impact of transformational leadership on the team is reflected not only in team members' opinions and perceptions, but also in relationships that may affect the team's performance. Team members are led to feel that their work is of great significance, which enhances their intrinsic motivation [31].

This intrinsic motivation is manifested in the following ways. Psychological empowerment increases the level of organisational commitment by enhancing members' attitudes to work and organisational citizenship, and increases the meaning of the work and members' self-efficacy, work autonomy, and work impact, and thus ultimately enhances team creativity [32,33]. In addition to promoting innovation, transformational leadership reduces the negative impact of heterogeneity on team creativity and promotes positive collusion and the exchange of ideas and knowledge [34,35]. In particular, transformational leadership encourages team members to adopt an open and inclusive

approach to different viewpoints and values, and to try to improve the motivation of their fellow team members to make full use of diverse cognitive resources.

The aim of the scientific research team is to solve complex social problems, and thus transformational leadership enables the selection of cross-discipline themes, attracting diverse knowledge and methods, promoting interdisciplinary cooperation, and ultimately improving team innovation performance [6]. The innovation process involves a high level of uncertainty, which presents significant psychological challenges for team members. The encouragement provided by transformational leaders ensures that team members are not afraid to explore their diversity, thereby reducing tension and creating a more relaxed environment in which they can discuss possible solutions, thus creating a more powerful psychological security system [8].

Transformational leadership can help build an organisational culture that displays inclusiveness toward members of other organisations [36]. Therefore, we propose the following hypotheses:

**Hypothesis 4.** *Transformational leadership positively moderates the relationship between disciplinary heterogeneity and innovation performance.*

**Hypothesis 5.** *Transformational leadership positively moderates the relationship between cognitive heterogeneity and innovation performance.*

**Hypothesis 6.** *Transformational leadership positively moderates the relationship between organisational heterogeneity and innovation performance.*

## 3. Material and Methods

### 3.1. Procedure and Sample

The selection of the sample was mainly based on two factors. First, the sample objects must have had experience in multidisciplinary or interdisciplinary collaboration and must have obtained relevant outcomes from interdisciplinary collaboration. Second, the result of estimating the sample object is as far as possible the innovation category.

Therefore, the humanities disciplines were excluded, after taking into account the positive relationship between innovative research and economic management disciplines, we confined the research sample to the fields of science and technology, and eventually chose the field of agricultural medicine. As the aim of this study is to analyze innovation by university scientific research teams, the person completing the questionnaire needs to have a comprehensive understanding of the overall innovation performance of the team, including annual funding, research results, research partners, and leadership style. Therefore, the aim of this investigation is to identify the leaders of the university scientific research team.

The survey was conducted over a seven-month period. We followed a third-stage sampling procedure. First, an English-language version of the questionnaire was prepared, and then this was translated from English to Chinese. Second, the questionnaire was independently tested using scientists to check its accuracy. Third, the sample firms were contacted either in person or by telephone or email to obtain their agreement to participate. Then the Chinese version and feedback were independently translated into English to ensure conceptual equivalence. We sent 870 questionnaires and collected data and obtained 305 responses with a 30.5% percent response rate. There was a wide discipline distribution of sample teams, including science (chemistry, physics, biology, etc.), engineering (chemical engineering, biological engineering, computing engineering, etc.), agronomy (agriculture, agricultural informatics, etc.); medicine (pharmaceutical, pharmacology, clinical medicine, etc.); economics (macroeconomics, microeconomics, industrial economics, etc.); management (marketing, management science and engineering, electronic commerce, etc.). Sample teams covered the east, south, west, and north parts of China.

In the quantitative study of heterogeneity, based on existing studies [37–39], considering the lack of more in-depth studies on the heterogeneity of implicit features in existing studies, these implicit dimensions may produce more stable and more significant influence and explanatory power on team innovation result variables. Therefore, although the study of implicit heterogeneity is more challenging than that of explicit heterogeneity, it enables the heterogeneity of the team to be subdivided into the following areas: disciplinary heterogeneity, cognitive heterogeneity, and organisational heterogeneity.

*3.2. Measures and Variables*

**Dependent and independent variables**. The survey of disciplinary heterogeneity was based on a proportion of a particular category, and then items were transformed to a seven-point Likert scale. The measure of disciplinary heterogeneity was assessed with five items and measured on a seven-point scale, ranging from 1"strongly disagree" to 7 "strongly agree." Also, we borrowed a combined method from van Der Vegt and Bunderson (2006) [40]; seven categories of disciplines include: science; engineering; agronomy; medicine; economics; management, and so on.

Organisation heterogeneity. This study draws on the organisation classification of research, and divides organisations into the following categories: universities, enterprises, research institutions, and other organisations.

Cognitive heterogeneity. Our measurements mainly refer to Shin's (2012) [34] scale of cognitive heterogeneity; seven items (Table 1) are included in the scales (Cronbach's $\alpha$ = 0.887).

**Table 1.** Items measuring cognitive heterogeneity.

| Variable | Items |
|---|---|
| Cognitive heterogeneity | There are differences in the way you think about problems.<br>There are differences in knowledge and technical backgrounds.<br>There are differences in task decisions.<br>There are differences in cognition of task influence factors.<br>There are differences in how you choose to complete a task.<br>There are differences in the world view.<br>There are differences in faith. |

**Moderating variables**. Transformational leadership. This study draws on eight items (Table 2) developed by Garcia-Morales, Llorens-Montes and Verdu-Jover (2008) [41] and the transformational leadership style scale of Song, Tsui, and Law (2009) [42] (Cronbach's $\alpha$ = 0.911).

**Table 2.** Items measuring transformational leadership.

| Variable | Items |
|---|---|
| Transformational leadership | Very capable, courageous and confident<br>Demonstrated determination in the process of accomplishing a goal<br>Make subordinates feel happy<br>For the benefit of the team, regardless of personal gains and losses<br>Express to subordinates their expectations of high performance<br>Talk passionately about the tasks that need to be done<br>Leaders to portray an inspiring future for all<br>To convey a sense of mission to all |

**Innovation performance.** The measurement of innovation performance mainly refers to She and Chen (2005) [43] when considering innovation results such as patent characteristics. Along with Hollingsworth (2007) [28], they point out that the process of scientific discovery is highly uncertain, and that most scientific discoveries take several years. The results of innovation projects generally accumulate over a long period of time, and so a measurement process is needed. Therefore, this study mainly considered innovation along two dimensions, an initiative dimension and an accumulation

dimension. The initiative dimension includes: initiative concept, initiative technology, open up a new field of research, and provide a new technology and research tools. The accumulation dimension includes: published SCI, SSCI and other high-level articles, and the results of the application of the invention patent and so on (Cronbach's $\alpha$ = 0.899).

### 3.3. Data Analysis Methods

The results show that there is significant internal consistency between the items measuring innovation performance. The innovation performance of the university is measured on a seven-point Likert-type scale, where 1 = "very low" and 7 = "very high." As a result of the traditional statistical regression method for scale, the method of calculating mean value is adopted. Considering the innovation evaluation method that is used, similar to other innovation performance evaluation methods using mean regression calculations, it is possible that the originality of the innovation cannot be highlighted. For example, there are two teams, team A and team B, which undertake innovation activity, and team A only produces one result, which is the gravitational waves of general relativity discovery, but achieves nothing else, that is, no published articles and no patents, the team's performance might only be rated as one "7." Meanwhile, the innovation of team B might have produced a number of results, such as a published article, improvements to previous research outcomes, and patent applications, even though team B's results did not represent a significant innovation, and thus team B's innovation performance might be rated as double "5" and one "6."

From the above description, the original performance of team A is more prominent than team B. Based on the above example, in evaluating innovation performance, we can no longer use the general innovation performance mean reversion calculation method, and instead choose the Poisson regression method, in which the measurement scale is converted so that a score of 7 is allocated a value of 1, while scores of 1–6 are allocated a value of 0.

## 4. Results

### 4.1. Descriptive Statistics

This study uses Pearson correlation analysis to describe and analyze all variables including control variables, independent variables, moderating variables, and dependent variables. The results are shown in Table 3. It can be seen that there is no multiple collinearity problem.

**Table 3.** Descriptive statistics.

|  | (1) | (2) | (3) | (4) | (5) | (6) | (7) | (8) |
|---|---|---|---|---|---|---|---|---|
| Team age | 1 | | | | | | | |
| Number of researchers | 0.399 ** | 1 | | | | | | |
| Annual funding | 0.299 ** | 0.421 ** | 1 | | | | | |
| Disciplinary heterogeneity | −0.101 | 0.003 | 0.016 | 1 | | | | |
| Cognitive heterogeneity | −0.143 * | −0.012 | −0.018 | 0.026 | 1 | | | |
| Organisational heterogeneity | 0.171 ** | 0.159 * | 0.251 ** | 0.280 ** | −0.067 | 1 | | |
| Transformational leadership | 0.013 | 0.115 | 0.096 | −0.055 | 0.107 | 0.014 | 1 | |
| Innovation performance | 0.130 * | 0.079 | 0.063 | 0.071 | 0.146 * | 0.080 | 0.287 ** | 1 |
| Mean | 3.07 | 2.38 | 3.19 | 0.525 | 0.335 | 0.690 | 5.560 | 2.333 |
| $\Delta R^2$ | 1.285 | 1.181 | 1.285 | 0.324 | 0.171 | 0.283 | 1.102 | 3.266 |

Note: ** $p < 0.01$, * $p < 0.05$.

The correlation coefficient between cognitive heterogeneity and innovation performance is 0.146 ($p < 0.05$), while that between transformational leadership style and innovation performance is 0.287 ($p < 0.01$). Correlation can only be used as preliminary evidence of a relationship between variables, and so for further validation of our models and research assumptions, we use Poisson regression analysis.

## 4.2. Relationship between Heterogeneity and Innovation Performance

In this study, Poisson regression analysis was used to test three independent variables: the relationship between disciplinary heterogeneity and innovation performance, cognitive heterogeneity and innovation performance, as well as organisational heterogeneity and innovation performance.

We propose that there is a positive correlation between heterogeneities and innovation performance (Hypotheses 1–3). Moreover, we propose that transformational leadership positively moderates the relationship between heterogeneities and innovation performance (Hypotheses 4–6). Prior to testing, variables were centralized to reduce the multicollinearity in the regression equation. To test H1–H3, three regression models were built as shown in Table 4. Model 1 focuses on the influence of control variables (team age, number of researchers, and annual funding). Model 1, which only includes the control variables, serves as the baseline model; in Model 2, three independent variables (disciplinary heterogeneity, cognitive heterogeneity and organisational heterogeneity) were entered into the equation to test the effect of team heterogeneities on innovation performance. In model 3, the cross term of heterogeneity and transformational leadership were entered into the equation to test the moderating effect proposed by Hypotheses 4, 5, and 6.

**Table 4.** Regression Analysis of the Relationship between Heterogeneity and Innovation Performance.

| | Dependent: Innovation Performance | | |
| --- | --- | --- | --- |
| | **Model 1** | **Model 2** | **Model 3** |
| *Control variables* | | | |
| Team age | 0.123 *** | 0.166 *** | 0.165 *** |
| | (0.0349) | (0.0358) | (.0360) |
| Number of researchers | 0.029 | −0.029 | −0.036 |
| | (0.0391) | (0.0416) | (0.0421) |
| Annual funding | 0.18 | −0.021 | 0.001 |
| | (0.0352) | (0.0366) | (0.0374) |
| *Independent variables* | | | |
| Disciplinary heterogeneity | | 0.111 * | 0.214 *** |
| | | (0.0460) | (0.0535) |
| Cognitive heterogeneity | | 0.180 *** | 0.181 *** |
| | | (0.0409) | (0.0479 |
| Organisational heterogeneity | | 0.087 † | 0.087 |
| | | (0.0494) | (0.0543) |
| *Moderating variables* | | | |
| Transformational leadership Z | | 0.497 *** | 0.530 *** |
| | | (0.538) | (0.0546) |
| *Cross terms* | | | |
| Disciplinary heterogeneity × Z | | | −0.292 |
| | | | (0.0614) |
| Cognitive heterogeneity × Z | | | 0.015 *** |
| | | | (0.0495) |
| Organisational heterogeneity × Z | | | 0.091 |
| | | | (0.0566) |
| Model statistics | | | |
| Likelihood Ratio Chi-Square | 20.683 | 162.653 | 185.501 |
| df | 3 | 7 | 10 |
| Sig. | 0.000 | 0.000 | 0.000 |

Note: *** $p < 0.001$; ** $p < 0.01$; * $p < 0.05$; † $p < 0.1$, two-sided.

As can be seen from Table 4, the two-tailed tests for Model 1, Model 2, and Model 3 are all significant ($p < 0.001$). In Model 2, cognitive heterogeneity corresponds to b = 0.180 ($p < 0.001$), and has a significant positive effect on innovation performance in colleges and universities. Thus, Hypothesis 2 is supported. Disciplinary heterogeneity corresponds to b = 0.111 ($p < 0.05$), and plays a significant role in

promoting innovation performance, thus Hypothesis 1 is supported. The organisational heterogeneity corresponds to b = 0.087 ($p < 0.1$), and has a significant positive effect on the innovation performance, thus Hypothesis 3 is supported. In summary, three independent variables (disciplinary heterogeneity, cognitive heterogeneity, and organisational heterogeneity) all had a significant positive effect on the team innovation performance.

Therefore, it can be assumed that Hypotheses 1, 2, and 3 are all supported. The results of Model 3 show that a transformative leadership style has a significant positive effect on the relationship between cognitive heterogeneity and innovation performance of b = 0.015 ($p < 0.001$ (Figure 1)). Further, organizational heterogeneity does not have a significant moderating effect on innovation performance. Thus, it is assumed that Hypothesis 5 is supported, and that Hypotheses 4 and 6 are not supported. Transformational leadership has a significant positive effect on cognitive heterogeneity and innovation performance, but there is no significant moderating effect on the relationship between organisational heterogeneity and innovation performance.

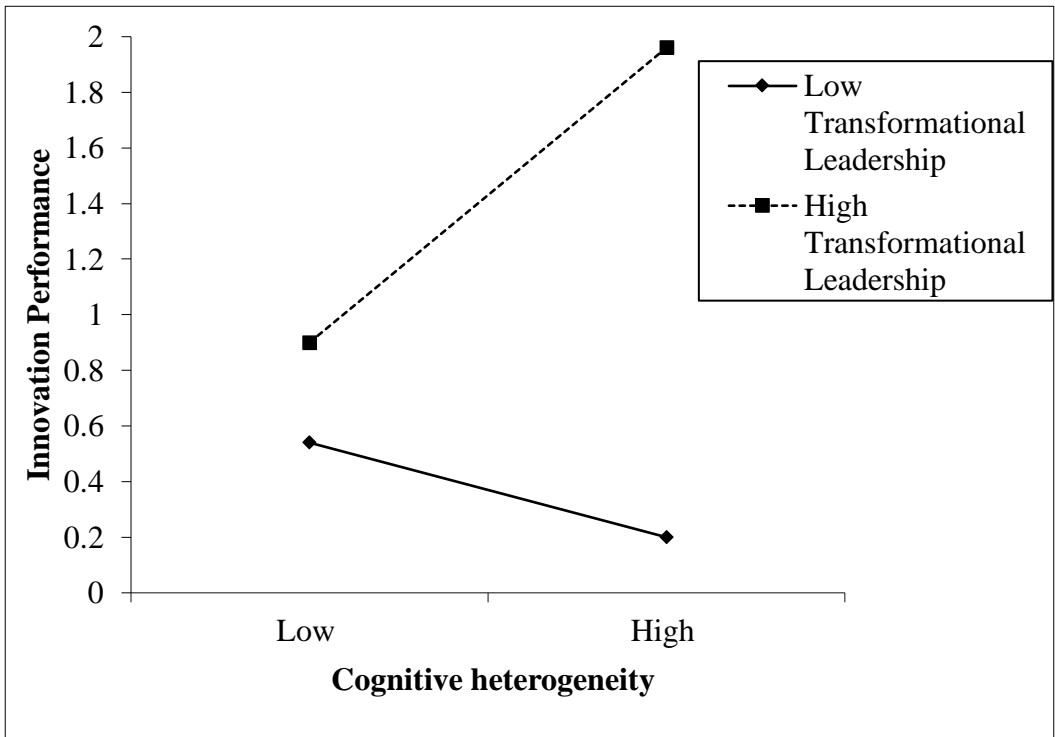

**Figure 1.** Transformational leadership and cognitive heterogeneity interaction for team innovation performance.

## 5. Discussion and Conclusions

### 5.1. Theoretical Implications

The theoretical contributions of this study are as follows.

Firstly, cognitive heterogeneity was introduced, and a direct measurement method using a psychological scale was adopted. We divided team heterogeneity into three categories: disciplinary heterogeneity, cognitive heterogeneity, and organisational heterogeneity. This is the first time that the cognitive dimension has been accorded the same degree of importance as the disciplinary and organisational dimensions. This provides a new perspective for the study of disciplinary heterogeneity in the future. The results showed that cognitive heterogeneity is positively correlated with innovation performance in universities, indicating that our hypotheses are supported.

Secondly, in the field of innovative research, heterogeneity measurements are generally based on alternative variables that measure the degree of heterogeneity indirectly [44,45]. This study uses a



scale to measure team members' heterogeneity directly, thereby avoiding the deviations that can occur with indirect measurement.

Thirdly, in the context of the university, the relationship between the heterogeneity and innovation performance is determined, and the importance of cognitive differences in promoting innovation is highlighted. Previous studies on heterogeneity have been relatively broad, and the theory on which it is based is complex, which has produced inconsistent results regarding the relationship between heterogeneity and innovation performance. Therefore, when studying innovation in the university, we cannot use these results as a reference point, and a new approach to research into the relationship between heterogeneity and innovation performance of universities is necessary. The results of the Poisson regression analysis showed that cognitive heterogeneity and organisational heterogeneity had significant positive effects on the innovation performance. This indicates that Hypotheses 1, 2, and 3 are supported. We confirm that there is a significant positive correlation between disciplinary heterogeneity and innovation performance, and our empirical results support the findings of previous studies [11,46]. Conversely, the results of Deutsch (2014) were refuted [10]. Hypothesis 2 confirms that our findings support those of Mitchell et al. (2017) [24] and Lavy, Bareli, & Ein-Dor (2015) [26], while refuting those of Williams and O'Reilly (1998) [22] and Ozuem & Sarsby (2014) [23]. Furthermore, some programmes can be employed to enhance trust and cooperation among interdisciplinary scientists, which includes sharing interdisciplinary research practices, sharing understanding, intensive interaction, questioning, and so on [46]. The results in relation to Hypothesis 3 are consistent with those of Walsh, Lee, and Nagaoka (2016) [19].

The most important thing is that, cooperation between teams from different organisations may also have a negative impact, but the benefits of the heterogeneous resources and knowledge derived from collaboration across organisations are more pronounced [20,47]. We know that the theoretical basis of heterogeneity promoting innovation performance is mainly related to the resource-based theory and social identity theory. Research teams in colleges and universities that are engaged in complex scientific research are faced with relatively difficult problems, the system is more complex, and team members from different disciplines are required to cooperate, even though cognitive heterogeneity may lead to temporary conflict between individuals. However, most of these researchers have lofty ideals and aspirations, and so the promotion of cooperation, conflict resolution, understanding, and tolerance strengthens their mutual understanding and team cohesion, thereby enhancing the team's ability to innovate [11]. In short, the positive effect of heterogeneity is greater than the conflict it creates.

Therefore, in the study of heterogeneity and innovation performance, the resource-based theory and social identity theory are strengthened and the view of conflict is weakened. This result shows the importance of boundary-spanning disciplines, which is consistent with the work of Whalen (2018) [48]. In addition, the application of leadership theory to innovation in the university is broadened. The results show that transformational leadership has a significant moderating effect on the relationship between cognitive heterogeneity and innovation performance, while it does not have a significant moderating effect on the relationship between discipline heterogeneity and organisational heterogeneity. In other words, cognitive heterogeneity promotes innovation performance in the context of a transformational leadership style. This means that transformational leaders can reduce the negative impact of heterogeneity on team creativity and promote the collision and exchange of ideas and knowledge. Transformational leadership encourages team members to adopt an open and inclusive approach to different viewpoints and values, thereby increasing the creative motivation of team members by encouraging them to utilize diverse cognitive resources [34,35]. Based on the moderating effect of transformational leadership on the relationship between cognitive heterogeneity and innovation performance, it can be seen that the transformational leadership style is not effective in relation to discipline heterogeneity, which is static, but plays a positive role in relation to cognitive heterogeneity, which is dynamic. Therefore, special attention must be paid to the application of the transformational leadership style to the cognition of team members.

Moreover, the influence of the transformational leadership style varies in relation to different types of heterogeneity. In addition, transformational leadership does not have a moderating effect on the relationship between organisational heterogeneity and innovation performance. This suggests that the influence of organisational heterogeneity on innovation performance is not moderated by the leadership style. In this regard, we believe that it is possible that interdisciplinary cooperation in relation to innovation in colleges and universities is mainly concentrated in similar types of organisations because cooperation between different organisations and different disciplines is very difficult to achieve, and is not something that the leadership style can play a decisive role in moderating.

Finally, a Poisson regression analysis of innovation performance was developed to highlight the characteristics of innovation performance. Innovation performance is traditionally measured using a seven-point Likert-type scale, and then uses hierarchical regression to carry on the average value to the seven-level scale, and takes the average to return. This regression method inevitably allows the conspicuous of primitive innovation to be erased, resulting in an average performance, which is not very effective in characterizing innovation features. In this study, Poisson regression analysis was used to convert questionnaire responses into binary form. A score of 7 was allocated a value of 1, and a score of 1–6 was allocated a value of 0. This is the pioneering contribution of this study, which provides a more accurate method of evaluating innovation performance.

### 5.2. Managerial Implications

In the context of China, paying increasing attention to innovation and basic research, this study analyzes the relationship between interdisciplinary team heterogeneity and innovation in universities using Poisson regression analysis. The results of this study show that disciplinary heterogeneity, cognitive heterogeneity, and organisational heterogeneity can improve the innovation performance of interdisciplinary teams, and that the impact of cognitive heterogeneity is most significant. The aim of this study is to explore the effect of heterogeneity on innovation performance by research teams in universities. Although the theoretical model produced a range of results, they provide some management implications for innovation by heterogeneous teams. This has practical significance for strategic decision-making and interdisciplinary cooperation in our universities.

The management of heterogeneous teams in universities needs to exploit the advantages offered by heterogeneity while avoiding the potentially negative effects. First, cooperation among interdisciplinary teams can significantly promote innovation performance. Second, to improve innovation performance, it is necessary to pay special attention to the important role of cognitive heterogeneity in the process of interdisciplinary cooperation. It is the cognitive activity of individual creative thinking that leads to the integration of heterogeneous subject knowledge and the creation of new knowledge. The cooperation moves from static knowledge integration to dynamic individual cognition, and then expands to group cognition, which enables the integration of explicit subject knowledge and the creative thinking network of the interdisciplinary team. Therefore, our research emphasizes not only the integration of knowledge between different disciplines, but also the cognitive dynamics brought about by cross-discipline activities. The role of cognitive heterogeneity needs to be incorporated into future talent training models in scientific research areas. The choice of the participants should pay special attention to their cognitive and communication abilities. The management of heterogeneity should emphasize creating an atmosphere of team harmony, thereby avoiding any prejudice and negative feelings among team members, which may affect the performance of creative tasks and cooperation between team members. Third, this study provides ideas for leaders' choices in relation to interdisciplinary cooperation. The tendency of leadership style depends on specific circumstances. For example, for interdisciplinary collaboration, it is possible that spontaneous processes are more desirable, and transformational leadership may have a negative impact on teamwork. However, team coordination at the cognitive level requires a transformational style of leadership. Our research on the effect of transformational leadership will help universities to choose the appropriate leadership style for interdisciplinary cooperation.

### 5.3. Future Directions and Limitations

The article opened with the observation that universities are increasingly seeking interdisciplinary collaboration to achieve innovation. We need to see this issue in the context of Chinese scientific areas, which pay heavy attention to the importance of leadership. This study benefited from the prior work of scholars in the fields of heterogeneity, leadership theory, and innovation. In response to the challenges facing China in relation to innovation, and drawing on resource-based theory, leadership theory, social cognitive theory, and social identity theory, we constructed a theoretical model of the relationship between interdisciplinary team heterogeneity and innovation performance in universities. Poisson regression analysis produced some interesting findings. However, there are some limitations to this study, which provide opportunities for future work.

Firstly, we selected science and technology disciplines, and ignored humanities and arts disciplines. Therefore, future studies could include the humanities and arts disciplines, which might provide some interesting results. Secondly, the small-scale organisations were consciously ignored in our study, we subjectively proposed that innovation is seldom relative to these organisations. Therefore, the findings may not represent all Chinese universities, which weakened the generality of the findings. Thirdly, in this study, heterogeneity was examined from three dimensions. However, this does not encompass all characteristics of team heterogeneity. There are several important factors that have not been included in the dimensions, such as institutional factor, strategic direction, cultural factor, and so on. In addition, we should focus on the research on effect of innovation policy [49,50]. Finally, the dynamic interactive relationships between disciplines, organisations, and cognition should be taken into account in future research.

**Author Contributions:** S.H. designed the study, interpreted data and wrote the paper. L.M. analyzed data. J.C. was the mentor and was responsible for leading the group. W.M. analysed data , polished and reviewed the paper.

**Funding:** This research is supported by the National Natural Science Foundation of China (Grant No. 71704155)

**Conflicts of Interest:** The authors declare no conflicts of interest.

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
