# Peer review of "The Effect of Heterogeneity and Leadership on Innovation Performance: Evidence from University Research Teams in China"

_sustainability, doi:10.3390/su11164441_

Round 1

Reviewer 1 Report

Abstract- Mention some of the findings in the abstract. More detail regarding the methodology can also be provided.

Hypotheses well motivated and designed.

Results and discussion well balanced and links to previous findings and studies.

The authors mentioned factor analysis in the abstract, this was not done in the final paper?

Overall a good and interesting paper.

Author Response

Letter of amendments

Dear Reviewers

Your comments and those of the reviewers were highly insightful and enabled us to greatly improve the quality of our manuscript. In the following pages are our point-by-point responses to each of the comments of the reviewers as well as your own comments.

Revisions in the text are shown using red highlight for additions, and strikethrough font [example] for deletions. In accordance with three reviewers’ suggestion, we made a careful revision. We hope that the revisions in the manuscript and our accompanying responses will be sufficient to make our manuscript suitable for publication in Sustainability.

Responses to the comments of Reviewer #1

Abstract- Mention some of the findings in the abstract. More detail regarding the methodology can also be provided.

Response: Thanks very much for your suggestion. The methodology has been added in the Abstract.(p. 1, lines 22-24. ).

The authors mentioned factor analysis in the abstract, this was not done in the final paper?

Response: Thanks very much for your suggestion. The description of factor analysis has been delected in the Abstract.

Reviewer 2 Report

The authors have made an econometric effort to measure aspects of innovation and this is commendable. In this sense, the methodology can be discussed little, although it seems to me that the work also has excessive features of abstraction. The authors focus preferentially on the sciences and forget other fields of knowledge. In fact, in its final words (lines 446 and following) it is when perhaps they explain more specifically what the reader really expected in the previous development of the article. The phenomena of innovation, in addition, require some comment that does not seem sufficiently collected in the article: 1. The role of the public sector, 2. The connection with the private sector or with the private sector plots allowed by the Chinese economy, 3. The transfer of knowledge from "public" research teams to private demands. In this regard, I recommend that the authors incorporate the most recent works of Mariana Mazucatto, who has deeply investigated these aspects related to the economics of innovation.

Author Response

Letter of amendments

Dear Reviewers

Your comments and those of the reviewers were highly insightful and enabled us to greatly improve the quality of our manuscript. In the following pages are our point-by-point responses to each of the comments of the reviewers as well as your own comments.

Revisions in the text are shown using red highlight for additions, and strikethrough font [example] for deletions. In accordance with three reviewers’ suggestion, we made a careful revision. We hope that the revisions in the manuscript and our accompanying responses will be sufficient to make our manuscript suitable for publication in Sustainability.

Responses to the comments of Reviewer #2

The authors focus preferentially on the sciences and forget other fields of knowledge. In fact, in its final words (lines 446 and following) it is when perhaps they explain more specifically what the reader really expected in the previous development of the article. The phenomena of innovation, in addition, require some comment that does not seem sufficiently collected in the article: 1. The role of the public sector, 2. The connection with the private sector or with the private sector plots allowed by the Chinese economy, 3. The transfer of knowledge from "public" research teams to private demands. In this regard, I recommend that the authors incorporate the most recent works of Mariana Mazucatto, who has deeply investigated these aspects related to the economics of innovation. With these improvements, the article could be published.

Response: Thanks very much for your suggestion. We modifed the content based on the works of Mariana Mazucatto, Reiter-Palmon and so on.

Reviewer 3 Report

The article provides interesting insights into exploring the relationship between team heterogeneity and innovation performance exemplified by the research teams in Chinas colleges and universities. 

In general, the structure of the article is correct and transparent presented, however, I would like to suggest an improvement by changing the heading of section into “5. Discussion and conclusions”.

Furthermore, the article would benefit from changing a bit the title. It seems to be not quite adequate to the theoretical description and empirical findings. I would suggest the following title: “The effect of heterogeneity and leadership on innovation performance: Evidence from university research teams in China”. That reformulation of the article title could improve academically adequate first impression and increase understanding of the content.

Moreover, in the title of the article you used the expression “original innovation”. What do you understand under this term? Is there an official classification available in the literature? Do you know non-original innovations? This is too colloquial reasoning, which unfortunately has not been explained in the article. Even in the section regarding the literature review there is not a single word on the subject but only in the Introduction (lines: 26-29) and in the section “Method” (lines: 262-271).

Further comments on individual parts:

The Abstract is not precise enough. The abstract should be improved by referring to a clear research objective, methods used as well as implications and added value.

The introduction should provide not only a research background but also an indication of a precise research objective and methods used. Adding a brief indication of the logic of presenting the research material (i.e. the brief description of the content of each section of the paper) in the last paragraph of the introduction is highly recommended.

The Literature review presents important highlights on the current state of the art. The authors justified precisely the main issues and their importance. However, it is recommended that the authors utilize more current literature.

The use of research methods is adequate, however the methodology requires better explanation, including the justification of sample selection, data collection and data analysis. It is necessary to describe the choice of research method, including its strengths and weaknesses for the research. It would be also important to clarify when the research study was conducted? In line 221 you mentioned that the survey was conducted over a seven-month period but the key questions are when and where it was carried out? What period is covered by the data in the research sample? Who participated in the survey? How the research sample was chosen? What was the right research subject? – the agriculture medicine (line 213) or science and technology (line 453). Please take care of the consistency of information and reliability because using different concepts and categories causes confusion. This section have to be improved and refined.

The result section is extensive and interesting, however, the quality of Figure 1 must be improved.

Discussion and conclusion section are well presented, including the most important results and implications.

Finally, the References must be largely complemented and adjusted to the editorial requirements of the Journal. Unfortunately, most of the  References in the text do not appear in the References list at the end of the article. Detailed instructions for references are presented on the website: https://www.mdpi.com/journal/sustainability/instructions#references

Furthermore, the following changes would be reasonable:

Line 7: Use “innovation” instead of “original innovation” – the same applies to the entire article Line 18: Use “Interdisciplinary cooperation” instead of „Interdisciplinary” / use “innovation performance” instead of “original innovation” / use team heterogeneity” instead of “cognitive heterogeneity” Line 19: remove the following keywords: “Poisson regression; Chinese sustainable development” - and insert “university”, “China” Line 21: The author (Boulding, 1976) is not listed in References - the same applies to other authors cited in the article Line 36: write “The resource-based theory” instead of “The Knowledge-based theory” Line 37: remove the word “improved” Line 38: use “cognitive” instead of “cognition” Line 59: write “The results outline” instead of “The Results section outlines” Line 85: use “cognitive” instead of “cognition” Line159: write “Organisations” instead of “organisations” Line 196: write “Knippenberg” instead of “knippenberg” Line 206: Use “Material and methods” instead of “Method” Line 236: write “Organisation” instead of “organisation” Line 241 (in the Table): write “Variable” instead of “Variables”. Moreover, use the same style for all items, i.e. it should be “There are differences in cognition of …” Line 242: insert a space between table 1 and text Line 245 (in the Table): write “Variable” instead of “Variables” Line 253: write “open up” instead of “Open up” Lines 256-274: here you could insert a new subsection "Data analysis methods" Line 288: write “Poisson” instead of “poisson” Line 297: write “In Model 2” instead of “in Model 2” Line 304: adjust the size of the letters Line 318: write “it can be assumed” instead of “assume” Line 327: write “Discussion and conclusions” instead of “Discussion” Line 391: write “Moreover” instead of “Also” Line 400: write “Poisson” instead of “poisson” Lines 448-449: there are listed different theories not mentioned before in the article (e.g. creativity theory), while the social cognitive theory was not indicated.

The academic language is correct, however the expressions, wording, style - might be improved. General proofreading would be advisable.

Author Response

Letter of amendments

Dear Reviewers

Your comments and those of the reviewers were highly insightful and enabled us to greatly improve the quality of our manuscript. In the following pages are our point-by-point responses to each of the comments of the reviewers as well as your own comments.

Revisions in the text are shown using red highlight for additions, and strikethrough font [example] for deletions. In accordance with three reviewers’ suggestion, we made a careful revision. We hope that the revisions in the manuscript and our accompanying responses will be sufficient to make our manuscript suitable for publication in Sustainability.

Responses to the comments of Reviewer #3

In general, the structure of the article is correct and transparent presented, however, I would like to suggest an improvement by changing the heading of section into “5. Discussion and conclusions”.

Response: Thanks very much for your suggestion. We have changed the heading of section “ 5. General Discussion” into “ 5. Discussion and Conclusions” . (p. 9, lines 381)

Furthermore, the article would benefit from changing a bit the title. It seems to be not quite adequate to the theoretical description and empirical findings. I would suggest the following title: “The effect of heterogeneity and leadership on innovation performance: Evidence from university research teams in China”. That reformulation of the article title could improve academically adequate first impression and increase understanding of the content.

Response: Thanks very much for your professional suggestion. We have changed title as “The effect of heterogeneity and leadership on innovation performance: Evidence from university research teams in China”.  (p. 1, lines 4)

Moreover, in the title of the article you used the expression “original innovation”. What do you understand under this term? Is there an official classification available in the literature? Do you know non-original innovations? This is too colloquial reasoning, which unfortunately has not been explained in the article. Even in the section regarding the literature review there is not a single word on the subject but only in the Introduction (lines: 26-29) and in the section “Method” (lines: 262-271).

Response: Thanks very much for your suggestion. The term original innovation may be more appropriate for the Chinese context than a universal academic concept. So “Original innovation” has been substituted by the word “innovation”.

Further comments on individual parts:The Abstract is not precise enough. The abstract should be improved by referring to a clear research objective, methods used as well as implications and added value.

Response: Thanks for your comments. The research objectives, methods, findings and implications of this study have been further clarified in the Abstract section. (p. 1, lines 23-35)

The introduction should provide not only a research background but also an indication of a precise research objective and methods used. Adding a brief indication of the logic of presenting the research material (i.e. the brief description of the content of each section of the paper) in the last paragraph of the introduction is highly recommended.

Response: Thanks for your kind comments. We have added one paragraph to indicate the the brief description of the content of each section. (p. 2, lines 94-99)

The Literature review presents important highlights on the current state of the art. The authors justified precisely the main issues and their importance. However, it is recommended that the authors utilize more current literature.

Response: Thanks for your comments. We have added a wide range of current literatures. All the literatures are listed in the section of References.

The use of research methods is adequate, however the methodology requires better explanation, including the justification of sample selection, data collection and data analysis. It is necessary to describe the choice of research method, including its strengths and weaknesses for the research. It would be also important to clarify when the research study was conducted? In line 221 you mentioned that the survey was conducted over a seven-month period but the key questions are when and where it was carried out? What period is covered by the data in the research sample? Who participated in the survey? How the research sample was chosen? What was the right research subject? – the agriculture medicine (line 213) or science and technology (line 453). Please take care of the consistency of information and reliability because using different concepts and categories causes confusion. This section have to be improved and refined.

Response: Thanks for your comments. We have deleted agricural medicine in lines 239. And we complemented key imformation about samples and methods.(p.6 lines 247-260).

The result section is extensive and interesting, however, the quality of Figure 1 must be improved.

Response: Thanks for your comments. We have Redrawn Figure 1. (p. 10, lines 369)

Discussion and conclusion section are well presented, including the most important results and implications.

Response: Thanks for your kind encouragement.

Finally, the References must be largely complemented and adjusted to the editorial requirements of the Journal. Unfortunately, most of the  References in the text do not appear in the References list at the end of the article. Detailed instructions for references are presented on the website: https://www.mdpi.com/journal/sustainability/instructions#references

Response: Thanks for your comments.The references have complemented and adjusted to the editorial requirements of the Journal.

Furthermore, the following changes would be reasonable: Line 7: Use “innovation” instead of “original innovation” – the same applies to the entire article

Response: Thanks for your careful and kind comments. We have used “innovation” instead of “original innovation” , which has been used as the sample applies to the entire article.

Line 18: Use “Interdisciplinary cooperation” instead of“Interdisciplinary” / use “innovation performance” instead of “original innovation” / use team heterogeneity” instead of “cognitive heterogeneity” Line 19: remove the following keywords: “Poisson regression; Chinese sustainable development” - and insert “university”, “China”

Response: We have revised the section of Keyword, and used “Interdisciplinary cooperation” instead of “Interdisciplinary” / use “innovation performance” instead of “original innovation” / use team heterogeneity” instead of “cognitive heterogeneity” ; removed the following keywords: “Poisson regression; Chinese sustainable development” - and insert“university”,“China” (p.1 lines 36-38).

Line 21: The author (Boulding, 1976) is not listed in References - the same applies to other authors cited in the article

Response: Thanks for your comments.The reference of Boulding (1976) have been complemented in the section of References.

Line 36: write “The resource-based theory” instead of “The Knowledge-based theory”

Response: Thanks for your comments. We used the term of “The resource-based theory” instead of “The Knowledge-based theory”. (p. 2, lines 55)

Line 37: remove the word “improved”

Response: Thanks for your comments. We have remove the word “improved” . (p. 2, lines 57)

Line 38: use “cognitive” instead of “cognition”

Response: Thanks for your comments. We have modified the sentence use “cognitive theory” instead of “cognition theory”. (p. 2, lines 57)

Line 59: write “The results outline” instead of “The Results section outlines” Line 85: use “cognitive” instead of “cognition”

Response: We have made revision as the reviewer’s comment. (p. 2, lines 79; p. 3, lines 100)

Line159: write “Organisations” instead of “organisations” ;Line 196: write “Knippenberg” instead of “knippenberg”

Response: We have made revision as the reviewer’s comment. (p. 4, lines 185; p. 5, lines 222)

Line 206: Use “Material and methods” instead of “Method” ; Line 236: write “Organisation” instead of “organisation”

Response: Thank the reviewer very much. We have used “Material and methods” instead of “Method” (p.5 lines 232); and writed “Organisation” instead of “organisation” (p. 6, lines 276)

Line 241 (in the Table): write “Variable” instead of “Variables”. Moreover, use the same style for all items, i.e. it should be “There are differences in cognition of …” Line 242: insert a space between table 1 and text

Response: Thank the reviewer very much. We have used “Variable” instead of “Variables” . And we have used a unified style in the table. (p. 6, lines 281)

We have inserted a space between table 1, 2 and text (p. 6, lines 282;p. 6, lines 287 )

Line 245 (in the Table): write “Variable” instead of “Variables”

Response: Thank the reviewer very much. We have used “Variable” instead of “Variables” .  (p. 6, lines 286)

Line 253: write “open up” instead of “Open up”

Response: Thank the reviewer very much. We have used  “open up” instead of “Open up”.  (p. 6, lines 295)

Lines 256-274: here you could insert a new subsection "Data analysis methods" Line 288: write “Poisson” instead of “poisson” ; Line 297: write “In Model 2” instead of “in Model 2”

Response: Thank you very much for your comment. We have inserted a new subsection "Data analysis methods" in p. 7 lines 298. We have writed “Poisson” instead of “poisson” in p. 8 lines 331; writed “In Model 2” instead of “in Model 2” in p. 9 lines 342.

Line 304: adjust the size of the letters

Response: Thank you very much for your comment. We have adusted the size of the letters of “two-sided” in p. 8 lines 356.

Line 318: write “it can be assumed” instead of “assume” ; Line 327: write “Discussion and conclusions” instead of “Discussion” Line 391: write “Moreover” instead of “Also” Line 400: write “Poisson” instead of “poisson”

Response: Thank you very much for your comment. We have written “it can be assumed” instead of “assume” (p. 10 lines 372); And we have used “Discussion and conclusions” instead of “Discussion” “assume” (p. 10 lines 381); We have written “Moreover” instead of “Also” (p. 10 lines 409); We have written “Poisson” instead of “poisson” (p. 11 lines 454)

Lines 448-449: there are listed different theories not mentioned before in the article (e.g. creativity theory), while the social cognitive theory was not indicated.

Response: Thank you very much for your comment. We have deleted the theory which was not mentioned before. (p. 13 lines 506) And we added the social cognitive theory in p.13 lines 506.

The academic language is correct, however the expressions, wording, style - might be improved. General proofreading would be advisable.

Response: Thank you very much for your comment. We have polished our expressions in our article.

Round 2

Reviewer 3 Report

Thank you for the revised manuscript of your paper and the significant improvements.

This is now ready for publication.

Author Response

Dear reviewer,

Thanks very much for your comments.